Quantifying morphological variation in the Castilleja pilosa species complex (Orobanchaceae)

http://orcid.org/0000-0002-3125-6266 Jacobs Sarah J. 1 2 3 sarahjjacobs@gmail.com
Herzog Sarah 1 2
Tank David C. 1 2 3
1 Stillinger Herbarium, University of Idaho , Moscow, ID , USA
2 Biological Sciences, University of Idaho , Moscow, ID , USA
3 Institute for Bioinformatics and Evolutionary Studies (IBEST), University of Idaho , Moscow, ID , USA
Sosa Victoria
Electronic publication date: 2019 Jun 20
Publication date: 2019
Volume: 7
Electronic Location ID: e7090
Received 2018 Nov 6; Accepted 2019 May 6
Copyright: © 2019 Jacobs et al.
Copyright year: 2019
Copyright holder: Jacobs et al.
License: This is an open access article distributed under the terms of the Creative Commons Attribution License, which permits unrestricted use, distribution, reproduction and adaptation in any medium and for any purpose provided that it is properly attributed. For attribution, the original author(s), title, publication source (PeerJ) and either DOI or URL of the article must be cited.
License URL: https://creativecommons.org/licenses/by/4.0/

Keywords: Castilleja, Classification, Morphology, Species delimitation, Taxonomy, Type specimen, Multivariate morphological trait space

Funding: National Science Foundation (NSF) DEB-1253463 and DEB-1502061 This work was supported by the following grants from the National Science Foundation (NSF): DEB-1253463 (David C. Tank) and DEB-1502061 (to David C. Tank for Sarah J. Jacobs). The funders had no role in study design, data collection and analysis, decision to publish, or preparation of the manuscript.

==============================
Robustly delimited species are of paramount importance, the identification of which relies on our ability to discern boundaries between one species and the next. This is not difficult to do when species are very distinct from one another. However, in recently evolved lineages where putative species may have relatively few diagnostic features (e.g., species complexes composed of very similar species, the boundaries between which are often unclear), defining species boundaries can be more challenging. Hence, the field of species delimitation has widely advocated the use of multiple lines of evidence to delimit species, particularly in species complexes. Excessive taxonomic confusion, often the result of species descriptions that shift through time (e.g., during revisionary work and regional treatments), can further complicate the search for diagnostic features in species complexes. Here, as a first step in robustly delimiting species boundaries, we quantify and describe morphological variation in the Castilleja pilosa species complex. We first infer the morphospace of the species complex and use fuzzy-clustering techniques to explore the morphological variation in the system. Next, we hypothesize the position of type specimens within that morphospace. In so doing, we aim to visualize the impact that regional treatments have had on the conceptualization of taxa through time. We find that there is limited morphological variation among members of this complex, and we determine that the morphological concept of these species have shifted through time and are no longer accurately represented by species descriptions.

Introduction

Because they provide the basis for the recognition of one of the primary units of biodiversity, the species, classifications are the cornerstone of the biodiversity sciences. As such, classifications are vital to our understanding of biodiversity and the process of speciation. Therefore, the careful and robust delimitation of species is imperative. Species delimitation relies on our ability to define boundaries between one population and the next. Historically, this has been done using morphological evidence (Sneath & Sokal, 1973), ecological evidence (Van Valen, 1976), and more recently, in light of technical and analytical advances, molecular evidence (Baum & Shaw, 1995). Each criterion has limitations for being widely applied across the tree of life (de Queiroz, 2007), and no one criterion has been universally applied to defining species boundaries (de Queiroz, 2009). Instead, there has been a movement to include multiple lines of evidence in the delimitation of species (Padial et al., 2010; Schlick-Steiner et al., 2010; Carstens et al., 2013; Dejaco et al., 2016; Freudenstein et al., 2016). In addition to describing newly discovered species, species delimitation methods are often applied to existing classifications where species boundaries are poorly defined and/or sample assignment to species is difficult (e.g., Barley et al., 2013; Giarla, Voss & Jansa, 2014). In these cases, species delimitation is used in a validation context (where taxonomic boundaries are validated—i.e., individuals are assigned to a group a priori (Ence & Carstens, 2010)) and attempts to clarify species boundaries and which lines of evidence (morphological, ecological, molecular) do and do not describe species.

Traditionally, species have been described and established, in part, by designating a type specimen. From the type collection and other specimens examined during species discovery, morphological, ecological, and distributional traits are used to create a species description, providing a central reference point or general conceptualization of the new species, around which some amount of variation occurs (Fig. 1). However, the characteristics of this variation (the amount, the direction, etc.) are not static, and additional collections assigned to a species can shift the conceptual boundaries of the species, in particular how this is applied on-the-ground. For example, regional and floristic studies can result in treatments and species descriptions that incorporate variation observed in the field on a local scale. Revisionary work, typically happening at a broader scale (e.g., the Flora of North America), often recognizes overlapping variation between similar species and synonymizes names where appropriate. As a result, there can be a shift of species boundaries and known variation. In essence, these shifts can inflate or deflate the taxonomic concept of an entity, sometimes greatly outside the realm of its original description. Taxonomic drift such as this can result in a species description that no longer represents the range of described variation for a species, and instead only represents a portion of that variation (Fig. 1).

Figure 1 Schematic representing the amount of variation attributed to a species through time.

Species “A” and “B” are described, anchored by type collections “A” and “B”, and including variation around the type (smaller, dark continuous lines). Species descriptions of “A” and “B” are updated in regional treatments and floristic studies to incorporate variation observed in the field (arrows and lighter, continuous lines). Revisionary work recognizes overlap in variation in species “A” and “B” and synonymizes species “B” with species “A” (dark, most inclusive continuous line).

Often, the species involved in these taxonomic fluctuations are characterized as species complexes (i.e., groups of species that are difficult to distinguish from one another), and are already known to have overlapping variation that is difficult to classify. The shifting of recognized and ascribed variation through time and across treatments can further increase the fuzziness of species boundaries, making the identification of unknown individuals (and therefore the usefulness of the classification) even more difficult. This is further complicated when an unknown comes from a geographic boundary (or, conversely, one that is widespread but has varieties that occur in geographically restricted areas), served by two or more regional or localized treatments that have varying interpretations of variation within a taxon. This requires choices to be made by the identifier in preferring one treatment to another when treatments are in conflict (e.g., one treatment recognizes varieties, while another does not). Cases such as these—species complexes with a great deal of taxonomic confusion—are good targets for robust species delimitation. By clarifying and determining which lines of evidence distinguish species, classifications can be updated to reflect more accurate estimates of species boundaries.

Recognizing species in the genus Castilleja Mutis ex L.f. (Orobanchaceae Vent.) is notoriously difficult, particularly in the field. These difficulties largely stem from a nearly continuous range of variation both within and across taxonomic boundaries (Cronquist et al., 1984). The source of this morphological continuity is likely a combination of the young age of the lineage, the widespread and highly variable instances of polyploidy, and interspecific gene flow when species co-occur (Heckard & Chuang, 1977; Tank & Olmstead, 2008; Tank, Egger & Olmstead, 2009). This means that most often the characters that diagnose species are slight and often overlapping.

A good example of these difficulties can be found in the Pilosae alliance. Composed of approximately eight taxa and several varieties (Castilleja arachnoidea Greenm., C. cinerea A. Gray, C. nana Eastw., C. pilosa (S. Watson) Rydb. (with three named varieties: var. pilosa, var. longispica (S. Watson) N.H. Holmgren, and var. steenensis (Pennell) N.H. Holmgren), C. praeterita Heckard & Bacig., C. rubida Piper, C. salsuginosa N.H. Holmgren, and C. schizotricha Greenm.), members of this group have tubular flowers with less-showy corollas, short beaks, and a pouchy, lower corolla lip that has somewhat petaloid teeth (Fig. 2B) (Cronquist et al., 1984; Hitchcock et al., 1984; Wetherwax, Chuang & Heckard, 2012). This morphological alliance is unique among most Castilleja alliances in that it is composed of perennial diploids, with no documented polyploidy (Heckard & Chuang, 1977).

Figure 2 Distribution of focal taxa and diagrams of species morphology.

Distribution of focal taxa (A) and diagrams of species morphology (B). Filled circles represent specimens used to estimate ranges only (accessed through regional databases Consortium of Pacific Northwest Herbaria (pnwherbaria.org); Southwest Environmental Information Network (SEINet; swbiodiversity.org); University and Jepson Herbaria Specimen Portal (webapps.cspace.berkeley.edu); New York Botanical Garden (NYBG; nybg.org); Rocky Mountain Herbarium (RM; rmh.uwyo.edu)); open circles are individuals measured in this study. Castilleja pilosa var. pilosa (blue), C. pilosa var. longispica (orange), C. pilosa var. steenensis (red), C. nana (yellow). Photo Credit: JM Egger. Map data ©2017 Google.

Within the alliance, two widespread species are especially difficult to distinguish, particularly in the field—C. pilosa and C. nana. These taxa compose the core members of the pilosa species complex. C. pilosa is composed of three taxonomically recognized varieties, distinguished primarily by geography, in addition to slight variations in a suite of morphological characters. Castilleja pilosa var. pilosa is found in the Sierra Nevada, north and east into Oregon; Castilleja pilosa (S. Watson) Rydb. var. steenensis (Pennell) N.H. Holmgren is endemic to the high ridges of Steens Mountain in southeastern Oregon; Castilleja pilosa (S. Watson) Rydb. var. longispica (A. Nelson) N.H. Holmgren occurs in the southern half of Idaho, east into western Wyoming and Montana, and has a disjunct population in northern Idaho. These varieties are distinguished by calyx length, herbage pubescence, elevation, and geographic position (Fig. 2A) (Cronquist et al., 1984). Castilleja nana occurs throughout the central and southern Sierra Nevada range of eastern California and extends eastward on the high ridges of Nevada’s basin and range topography. Castilleja nana is primarily distinguished from C. pilosa by elevation, occurring between 2,400 and 4,200 meters, while C. pilosa is found primarily at lower elevations between 1,200 and 3,400 meters. Additionally, C. nana is often a smaller plant with decumbent branches and smaller features (Fig. 2A).

When C. pilosa and C. nana occur in sympatry and at the same elevation, it is often quite difficult to distinguish the two species. Additionally, many of the members of this complex occur across geographic and political boundaries and are represented in multiple, overlapping regional and floristic treatments (Cronquist et al., 1984; Hitchcock et al., 1984; Wetherwax, Chuang & Heckard, 2012). Subsequently, there has been a great deal of taxonomic confusion, demonstrated by the number of synonyms associated with C. pilosa and C. nana. Several of these incorporations are based on collections made in the Sierra Nevada where both species occur in sympatry, as well as northern California at the border with Oregon and Nevada. These regions also lie at the boundary between the Great Basin, the Pacific Northwest, and the Sierra Nevada and California floristic province, where a great deal of taxonomic work has been done. The taxonomic confusion in this group could be the result of any of the following factors: the young age of the lineage, the propensity for gene flow when species are sympatric, little to no morphological distinction between species, and/or the absence of species in the complex (i.e., the entire complex is actually a single lineage). As such, this complex is in great need for robust species delimitation.

Here we begin this process by quantifying morphological variation in the complex and assessing its correlation (or not) with the current taxonomy. By sampling many populations across the known ranges of these entities, identifying them using regional treatments, and measuring and analyzing a suite of morphological traits, we test the assumption that there are morphological clusters that correspond to taxonomic entities. We perform principal coordinate analyses to understand the position of individuals in morphospace, and then apply a non-hierarchical clustering method to assess the signal of morphological similarity that exists among these entities. In this way, we aim to quantify and begin to characterize the morphological variation in this species complex, information that will ultimately become part of a robust delimitation of species boundaries in this group.

Materials and Methods

Sampling and range estimation

Both mounted and unmounted collections of C. pilosa var. pilosa, C. pilosa var. longispica, C. pilosa var. steenensis, and C. nana were examined for this study, with emphasis placed on representing the hypothesized distributional ranges of these taxa. Prior to measurement, all collections were identified using the primary literature currently available—regional floras and treatments ((Wetherwax, Chuang & Heckard, 2012) California; (Hitchcock et al., 1984) Oregon, Idaho, and adjacent Montana and Wyoming; (Cronquist et al., 1984) Great Basin). Species ranges were estimated based on loan material and specimen label data accessed through regional databases (Consortium of Pacific Northwest Herbaria (pnwherbaria.org); Southwest Environmental Information Network (SEINet; swbiodiversity.org); University and Jepson Herbaria Specimen Portal (webapps.cspace.berkeley.edu); New York Botanical Garden (NYBG; nybg.org); Rocky Mountain Herbarium (RM; rmh.uwyo.edu)). Latitude and longitude were taken directly from collection labels, when available. In some cases, coordinates were not provided on the collection label, in which case they were estimated by hand based on locality information provided by the collector. For specimens whose identification we did not confirm (i.e., specimens not on loan), we only considered collections or identifications determined by collectors that we considered to have extensive expertise in Castilleja identification. All sampling information, including coordinates and voucher locations, can be found in Table S1.

Morphological measurements

We used a combination of continuous and categorical traits to characterize morphology. These traits are known to be taxonomically informative, and are widely used to identify and distinguish Castilleja species (Cronquist et al., 1984; Hitchcock et al., 1984; Chuang & Heckard, 1991; Hersch-Green & Cronn, 2009) (Table 1). Specimens were chosen for data collection based on the overall condition of the collection and maturity of the plant when it was collected, preferring specimens as close to peak flower as possible. Multiple stems within each collection were measured in order to record a complete set of measurements for each collection. Floral measurements were taken from dissected flowers rehydrated with Pohl’s (1965) solution. Flowers at peak maturity were identified, removed from the indeterminate inflorescence, and saturated with Pohl’s solution for five minutes. The bract, calyx, and corolla were separated from one another, and measurements taken from the dissected tissues (Fig. 2B). Habit, inflorescence, and leaf characters were taken from the specimen without further dissection; surface textures were taken from stem midway between the inflorescence and the base of the plant.

Table 1 Morphological characters measured in Castilleja.

	Character	Data type	Unit/level of measure	
Habit	
1	Plant height	C	cm	
2	Decumbent at base	N	3	
Surface textures	
3	Length of herbage pubescence	N	3	
4	Recurved hairs present	D	2	
5	Glandular hairs present	D	2	
Inflorescence	
6	Number of racemes per stem	C	8	
7	Length of raceme	C	mm	
Leaf	
8	Length of leaf	C	mm	
9	Width of leaf	C	mm	
10	Leaf lobing	D	2	
Bract	
11	Length of bract	C	mm	
12	Width of bract	C	mm	
13	Number of secondary lobe pairs	C	4	
14	Point of lobe attachment	C	mm	
Calyx	
15	Length of calyx	C	mm	
16	Tip of calyx to sinus 1	C	mm*	
17	Tip of calyx to sinus 2	C	mm*	
18	Calyx lobe subequality	C	mm; |#16–#17|	
19	Shape of tip of calyx segments	N	5	
Corolla	
20	Total length	C	mm	
21	Teeth to bottom of corolla	C	mm*	
22	Sinus of beak and lower lip to bottom	C	mm*	
23	Tube length	C	mm*	
24	Lower lip pouchy	D	2*	
25	Teeth petaloid	D	2*	
26	Stigmas exserted	D	2*	
27	Length of beak	C	mm; #20–#22*	
28	Beak length to tube length ratio	C	ratio; #20/#27	
Note:

Morphological characters measured in Castilleja. The first column following the character name reflects the type of character measured: continuous (C), nominal (N), or dichotomous (D); the second column provides the unit of measurement, the number of levels for nominal or ordinal data, and (when necessary) the formula for character calculation. Asterisks (*) indicate characters were not directly included in analyses, but used to calculate composite variables.

Nineteen continuous characters were measured from a total of 171 collections: C. nana (n = 50), C. pilosa var. longispica (n = 34), C. pilosa var. pilosa (n = 76), and C. pilosa var. steenensis (n = 11). Several continuous variables were used in the auto-calculation of additional continuous variables, thus creating a composite variable (Table 1, characters 18, 27, and 28). To avoid pseudo-replication of traits in the dataset, we removed the component traits (Table 1, characters 16, 17, 21, 22, 23, and 27), leaving only the composite variables in the dataset, resulting in thirteen quantitative characters. Nine categorical characters were recorded from the same 171 collections (Table 1). Three of these characters did not vary across individuals and were removed from the dataset (Table 1, characters 24, 25, and 26), leaving a total of six qualitative characters included in the analyses.

Data preparation and quantification of morphological variation

When present, raw measurements from different stems of the same collection were combined to produce an average measurement for each individual for each trait examined. Individuals with missing data for any of the traits measured (indicating the tissue was unavailable for sampling, a total of 60 collections) were removed from downstream analyses. We identified possible outliers in the dataset by calculating the multivariate normal density function of all continuous variables using the stats package in R (R Core Team, 2017), resulting in the pruning of 10 collections. Continuous variables were log transformed, and presence/absence data were coded as binary variables.

Principal coordinate analysis

To represent the morphological similarity in our dataset, we applied a metric, multidimensional scaling approach that positions each individual in a reduced dimension morphospace, preserving the distance relationship between individuals as well as possible (Gower, 1966). Because the categorical variables that we measured are taxonomically diagnostic, it was important to include them in a quantification of morphospace in this species complex. We performed a principal coordinate analysis (PCoA), which can handle both quantitative and qualitative data by using measures of (dis)similarity calculated from mixed variables (Gower, 1966; Legendre & Legendre, 1998). We calculated a dissimilarity matrix based on our log-transformed continuous variables, our nominal categorical variables, and our symmetric dichotomous variables, using Gower’s dissimilarity coefficient (Gower, 1971), as implemented using the daisy function in the R package cluster (Maechler et al., 2017). We then performed PCoA on the dissimilarity matrix using the function pcoa in the R package ape (Paradis, Claude & Strimmer, 2004). Principal coordinate analysis can sometimes result in negative eigenvalues when dealing with non-Euclidean distance measures (as we are doing here). As such, we used the Cailliez correction (Cailliez, 1983), where a constant is added to each original measure of dissimilarity (except the diagonals). Because PCoA is based on a pairwise distance matrix, there are approximately as many dimensions as there are pairwise comparisons, and they are ordered by their eigenvalues. By plotting each individual at the first two to three principal coordinates, we can represent the best possible Euclidean approximation of the morphological distance between them (Gower, 1982).

Fuzzy clustering

To explore and describe the signal of morphological similarity that we have quantified, we apply a clustering technique that can accommodate situations where cluster boundaries may not be clear-cut. Fuzzy clustering (Dunn, 1976; Kaufman & Rousseeuw, 2005) is a “soft” approach to clustering where individuals are assigned a probability of membership (the coefficient of membership) to each recovered cluster; this is in contrast to “hard” clustering where an individual is assigned to a single cluster only. The benefit of this type of clustering approach is that it can accommodate ambiguity in cluster assignments and provide more detailed information about the structure of the dataset.

The objective of the fuzzy-clustering algorithm is to minimize the within-cluster variance and maximize between cluster variance; put another way, the objective is to minimize the distance between two objects belonging to the same cluster. This is accomplished through an iterative procedure where cluster membership is initiated and a coefficient of membership is calculated for each individual based on the distance of the individual to the centroid of each cluster. The process is repeated until new clustering iterations fail to maximize the objective. After clustering, a final coefficient of membership to each cluster is calculated for each individual. When an individual is assigned equal coefficients to all clusters, it is described as having “complete fuzziness,” and can be imagined as falling in the “middle ground” between all clusters; when an individual has a membership close to 1 to a particular cluster, the clustering is essentially hard (i.e., it is a partition). Dunn’s (1976) normalized partition coefficient can be used to describe the overall fuzziness of an analysis, regardless of the number of clusters considered, where values close to 0 indicate high levels of fuzziness (near equal membership to all clusters), and values close to 1 indicate very low levels of fuzziness (i.e., hard partitions). After generating the coefficients of membership, one can find the hard partitioning scheme that most closely approximates the fuzzy clustering by assigning each individual to the cluster in which it has the largest membership.

One way to visualize the results of fuzzy clustering is by examining silhouette plots of the hard clusters. These plots are constructed of horizontal bars representing the silhouette coefficient (s(i)—a measure of that individual’s similarity to other members of the same cluster) of each individual in the analysis, organized by hard cluster assignment. When s(i) is at its largest for an individual (close to 1), that means that the individual is much more similar to other members of its cluster than it is to individuals outside of the cluster. When s(i) is low for an individual (closer to 0), it means that the individual is equally similar to both members of its cluster and members of other clusters. When an individual has an s(i) value that is negative, the within-cluster similarity is much smaller than the between cluster similarity. Finally, we can calculate the mean silhouette coefficient (i.e., the mean silhouette coefficient of all samples in the analysis) as a way of interpreting and validating the clustering. Kaufman & Rousseeuw (2005) suggest that datasets with silhouette coefficients less than or equal to 0.25 have no substantial structure, values between 0.26 and 0.50 indicate weak structure that could be artificial and require additional methods to corroborate, values between 0.51 and 0.70 suggest reasonable structure, and values between 0.71 and 1.0 suggest strong structure has been found.

Fuzzy-clustering analyses were run using the function fanny in the R package cluster (Maechler et al., 2017), and the same dissimilarity matrix for fuzzy clustering used for PCoA. Fuzzy clustering requires the user to define the number of clusters (k) to optimize. We chose to examine clustering of k = 4, 3, and 2 clusters. We begin at four because this corresponds with the number of named taxonomic entities focal to this study; three and two clusters were also examined to explore the morphological signature of the data. We further examined the effect of the membership exponent (a variable in the cluster optimization process) on our clustering results. It has been shown that higher values (near two) lead to greater fuzziness while lower values (near one) yield less fuzzy clustering (Kaufman & Rousseeuw, 2005). We examined the effect of this variable on clustering results by adjusting its value between 1.1 and 1.7, by increments of 0.1. We ran all fuzzy-clustering analyses for 100,000 iterations, to assure convergence.

Estimating position of type specimen in morphospace

To explore the position of type specimens in morphospace, we took the geographic position of each type specimen and found the nearest population of the same species from which we took morphological measurements. We make the assumption that these populations would have similar morphologies.

Results

Sampling

A total of 171 individuals were examined for this study. While normality is not a strict assumption of the approaches used here, extremely non-normal traits may affect results in unpredictable ways. As a conservative measure, we eliminated from downstream analyses approximately the top 10% of individuals that deviated extremely from the natural variability in the data. The impact of outlier removal on downstream analyses was examined and found to have minimal influence (Fig. S1). After data cleaning and outlier removal, our final dataset consisted of C. nana (n = 29), C. pilosa var. longispica (n = 23), C. pilosa var. pilosa (n = 52), and C. pilosa var. steenensis (n = 4), and covered the known ranges of each focal taxon (Fig. 2, open circles). Individuals measured, the herbarium housing each collection, and associated voucher information is available in the Supplemental Information, as well as raw data and associated analytical scripts (Table S1; Dataset S1, Dataset S2, respectively).

Quantifying morphological variation

Violin dot plots of individual quantitative trait values, grouped by taxonomic identity revealed a great deal of overlap in raw trait values for each taxon across many traits. In some cases, this overlap occurs across all focal taxa, as in bract width and leaf width (Fig. 3), where all taxa have widely overlapping trait values. In other cases, the distribution of trait values distinguishes one of the focal taxa from the remaining three. For example, C. pilosa var. steenensis has a larger beak to tube ratio than the remaining taxa (meaning that the difference in length between the tube and the beak is greater), C. nana has a longer bract than all varieties of C. pilosa and most C. pilosa var. longispica have shorter calyces than other varieties of C. pilosa and C. nana. There are also cases of interspecific overlapping trait distributions, as in plant height where C. nana and C. pilosa var. steenensis are generally shorter in height than C. pilosa var. longispica and C. pilosa var pilosa. We see a similar pattern of overlap in traits across taxa in our qualitative data. With the exception of the decumbent habit, no one qualitative trait is found primarily in one taxon, let alone exclusively (Fig. 4). In general, pubescence traits were equally variable across taxa, C. nana was the only taxon that occasionally lacked lobes on the leaves, and C. pilosa var. pilosa and C. pilosa var. steenensis were the only focal taxa that were never scored as having broader, deltoid shaped calyx lobes. Summary statistics for raw values of continuous traits and raw counts of categorical traits can be found in the supplement (Tables S2 and S3, respectively).

Figure 3 Raw trait values for continuous traits.

Violin dot plots of raw trait values (A–K) for the continuous traits measured in this study. C. pilosa var. longispica (orange), C. pilosa var. pilosa (blue), C. pilosa var. steenensis (red), and C. nana (yellow).

Figure 4 Summary of counts for categorical characters.

Summary of counts for categorical characters (A–F) measured here. Columns represent focal taxa whose area represents all individuals identified to that taxon in our dataset. Shading represents different character states scored for each individual. Dashes represent a character state unobserved in a particular taxon. For calyx lobe shapes, numbers are used in place of trait descriptions for simplicity. These correspond to: 1) linear, 2) lanceolate/linear, 3) lanceolate, 4) deltoid/lanceolate, and 5) deltoid.

Principal coordinate analysis

A Cailliez correction, equal to D′ = −0.5 * (D + 0.57237)2, was applied to all negative eigenvalues. The position of each individual in the first two and three principal coordinates are shown in Fig. 5, with 95% confidence ellipses around the mean position of each focal taxon in morphospace. An examination of axes 1 through 106 does not change interpretation of the results presented here; the first two principal coordinate axes represent the maximum morphological distance among individuals sampled, and the third axis reveals no further distinction (Fig. 5).

Figure 5 Results of Principle Coordinate Analysis.

Results of Principle Coordinate Analysis (PCoA) considering the first two axes of variation (A) and including a third axis (B). Individuals are represented by points in morphospace, and colored according to species identification: Castilleja pilosa var. pilosa (blue), C. pilosa var. longispica (orange), C. pilosa var. steenensis (red), and C. nana (yellow).

In general, and considering all three principal coordinate axes, individuals identified as C. nana (yellow) occupy a different part of the scatterplot than those identified as C. pilosa, including its named varieties (blue (var. pilosa), orange (var. longispica), and red (var. steenensis)). Considering only those individuals identified as C. pilosa, there is a large amount of overlap with no discernible position in morphospace unique to any variety (Fig. 5). Confidence ellipses lend support to this conclusion and further suggests a greater distinction of C. pilosa var. steenensis (in red) from any other focal taxon. The variation in distances of these individuals lies along a different axis than the rest of the focal taxa; however, the effect of sample size (n = 4) cannot be discounted.

Fuzzy clustering

We performed seven fuzzy-clustering analyses (corresponding to seven different values of the membership exponent variable; values between 1.1 and 1.7, in increments of 0.1) for each of three possible numbers of clusters (k = 4, 3, and 2). Different values of the membership exponent produced consistent results within each “k = X number” of clusters. For simplicity, we present the results from all clustering scenarios with a membership exponent of 1.3.

Fuzzy-clustering analyses, regardless of the number of clusters considered, resulted in clusters with small silhouette coefficients (both within and across clusters), and low values for the normalized Dunn coefficient (Fig. 6; Table 2). As cluster number was reduced, there appeared to be some small improvement in these measures (average silhouette coefficient increased from 0.2 (k = 4) to 0.22 (k = 3), and to 0.25 (k = 2) and normalized Dunn coefficient increased from 0.37 (k = 4) to 0.38 (k = 3), and 0.44 (k = 2)); however, overall these values are extremely low. Generally speaking, regions of overlap in morphospace coincide with lower probabilities of membership of each individual to each cluster (i.e., the probability of membership to each cluster was higher, as opposed to having an overwhelming probability of membership to any one single cluster), consistent across all clustering scenarios (Fig. S2).

Figure 6 Results of fuzzy clustering.

Results of fuzzy clustering for k = 4 clusters (A), k = 3 clusters (B), and k = 2 clusters (C). For each set of silhouettes, the width of each bar corresponds to the silhouette coefficient for that individual in the analysis. The silhouette width is a measure of that individual’s similarity to other members of the same cluster—when large (close to 1), that means that the individual is much more similar to other members of its cluster than it is to individuals outside of the cluster; when low (closer to 0), it means that the individual is equally similar to both members of its cluster and members of other clusters; when negative, the within cluster similarity is much smaller than the between cluster similarity. We also report the average silhouette coefficient for each analysis (k = 4, 3, 2). Bars are painted with colors corresponding to species identification and numbered for cross-referencing against Supplemental Table 4: Castilleja pilosa var. pilosa (blue), C. pilosa var. longispica (orange), C. pilosa var. steenensis (red), and C. nana (yellow).

Table 2 Results of fuzzy clustering analyses.

	k = 4	k = 3	k = 2	
	n	Avg s(i)	stdev s(i)	n	Avg s(i)	stdev s(i)	n	Avg s(i)	stdev s(i)	
Cluster 1	26	0.24	0.11	32	0.2	0.11	57	0.23	0.13	
Cluster 2	24	0.28	0.11	47	0.21	0.08	51	0.27	0.1	
Cluster 3	31	0.21	0.11	29	0.24	0.11				
Cluster 4	27	0.1	0.09							
Avg s(i) across analysis		0.2			0.22			0.25		
Normalized Dunn Coefficient		0.3768			0.3879			0.4424		
Note:

Results of fuzzy clustering analyses with k = 4, 3, and 2 clusters. Here we report average silhouette coefficients within and across clusters in analyses, as well as normalized Dunn coefficients for each analysis. Silhouette coefficients close to 0 represent less similarity, those close to 1 represent high similarity, and negative silhouette coefficients indicate likely misassignment to a cluster. The normalized Dunn coefficient is a measure of the overall fuzziness of an analysis. Values close to 0 indicate high levels of fuzziness (near equal membership of individuals to all clusters) and values close to 1 indicate very low levels of fuzziness (i.e., hard partitions).

A somewhat subjective approach to quantifying the structure in a dataset is to calculate the silhouette coefficient (SC) of the dataset (Kaufman & Rousseeuw, 2005). This value is the maximum, average silhouette coefficient of all possible numbers of clusters, from k = 2 as a minimum, to k = n as a maximum (n = 108, in this study). At k = 53, our standard 100,000 iterations of clustering were not enough to satisfy fuzzy-clustering objectives, and we ran into convergence issues. However, considering k = 2 through k = 53 clusters, the average silhouette coefficients were highest at k = 2 (average s(i) = 0.25), and steadily dropped as values of k increased.

To visualize the taxonomic composition of clusters, we painted the silhouettes with colors corresponding to the taxonomic identity of each individual. Across all three clustering schemes, one cluster is consistently composed of mostly C. nana individuals, with the remaining clusters being variously composed of all three varieties of C. pilosa. When we restrict the cluster number to two, the C. nana cluster begins to be more heavily composed of C. pilosa individuals (Fig. 6).

Discussion

Classifications are useful when they organize objects based on relationships, when they reflect similarities and differences among the constituent parts, and when they aid in the identification and placement of unknowns within the classification (Sokal, 1974; de Queiroz & Donoghue, 2011; de Queiroz & Donoghue, 2013). The species description, based in part on the type specimen, plays an important role in the creation and implementation of classifications, but with a reliance on it comes the challenge of tracing and managing type collections and species descriptions through time—a problem that we are still dealing with (Hitchcock, 1905; Dayrat, 2005). In addition, when objects are discrete and discontinuous, classifications are easy to build and use; however, when there is continuous variation in characters used in the classification, this becomes more difficult.

In this study we have closely examined morphology—a commonly used character for describing taxonomic boundaries—for four named taxa, from across their ranges, in a species complex known to be taxonomically difficult to diagnose. Here we have quantified a great deal of overlap in raw character traits that are typically used to diagnose species (when using taxonomic keys) in Castilleja (Figs. 3 and 4). In some cases, these traits are continuous across taxonomic boundaries (Fig. 3), emphasizing the extreme morphological similarity among these named entities. We continue to see little distinction among current taxonomic groups when we examine the morphospace described by all morphological characters that we measure here (Fig. 5). For example, C. pilosa, where we are essentially incapable of distinguishing taxonomic varieties using morphology alone (Fig. 5), even in C. pilosa var. steenensis, considered the most distinctive of the three varieties due to its isolation on Steens Mountain in SE Oregon (Hitchcock et al., 1984). Finally, when we interrogate morphospace for evidence of structure, we find little support (low silhouette widths for each cluster and low average silhouette widths for each clustering scenario, Fig. 6) and equal assignment probabilities of individuals that occur in areas of overlap to each cluster (Fig. S2).

And yet, despite the overall high levels of similarity we observe some consistent distinction between individuals of C. nana and C. pilosa, indicating some morphological distinction between taxa (Fig. 5). This is also supported by the results of fuzzy-clustering analyses that, regardless of the number of clusters considered, recover a cluster composed primarily of C. nana, with C. pilosa individuals variously scattered among the remaining clusters (Fig. 6). Several continuous traits distinguish C. nana from C. pilosa (Fig. 3; see also Fig. S3), however, the overlapping tails of these distributions, and the nature of these distinguishing traits (i.e., size and length traits that could be environmentally plastic), goes a long way towards explaining the morphological confusion that has plagued this complex historically.

It is clear that geographic and ecological characters must have played a dominant role in shaping the species descriptions in this complex. This is apparent from the species descriptions included both in regional and genus-wide treatments (Cronquist et al., 1984; Hitchcock et al., 1984; Wetherwax, Chuang & Heckard, 2012), as well as the inferred species ranges (Fig. 2). For example, C. nana does not occur in the northern limits of the C. pilosa range. So, if you encounter a relatively small individual in Idaho, there is no way to confuse it with C. nana (a California and Nevada species), as the ranges do not overlap, and the regional treatment does not consider C. nana (Hitchcock et al., 1984). Similarly, C. pilosa var. steenensis only occurs on Steens Mountain in Eastern Oregon. If you found a relatively small individual in central Oregon, you could only classify it as C. pilosa var. pilosa, using these regional treatments.

When species occur sympatrically, however, the distinction between named entities becomes much more difficult to parse. In the Sierra Nevada, C. pilosa var. pilosa (a moderate elevation taxon) and C. nana (a high elevation taxon) can co-occur at the limits of their elevational ranges (high and low, respectively) where environments are heterogeneous. Similarly, C. pilosa var. pilosa and C. pilosa var. steenensis can co-occur on the western slopes of Steens Mountain in the transition area between the high, exposed ridge and the surrounding lower elevation steppe. In heterogeneous habitats and at ecological boundaries, phenotypes can be accentuated and variable (van Kleunen & Fischer, 2005), potentially in response to local microhabitat conditions such as light availability and precipitation (Schlichting, 1986; Dorn, Pyle & Schmitt, 2000; van Kleunen, Fischer & Schmid, 2000; Nicotra et al., 2010). As a result, it is possible that in these areas of sympatry that correspond with environmental transitions, individuals could experience extreme conditions that may affect the morphological traits that we examine when we try to identify unknowns. We see this in several individuals from the Sierra Nevada that have extreme values in the traits that distinguish C. nana and C. pilosa (Fig. 7). Furthermore, these are the individuals that occur in the region of overlap in morphospace between these two taxa (Fig. 7).

Figure 7 Position of individuals in morphospace and geographic space.

Position of individuals with extreme trait values (A) in morphospace (C) and in geographic space (B). Individuals are color-coded according to taxonomic identification: Castilleja pilosa var. pilosa (blue), C. pilosa var. longispica (orange), C. pilosa var. steenensis (red), and C. nana (yellow). Histograms at the top of the diagram show trait distributions for C. nana (yellow) and C. pilosa (including all varieties, blue) for simplification. Vertical lines represent raw trait values and are color-coded corresponding to taxonomic identification.

In some cases, these regions of sympatry also correspond with hotspots of taxonomic synonymy historically—i.e., these sympatric areas are places where synonyms of currently accepted taxa were described (Fig. 8). For example, the area surrounding Lake Tahoe has seen the description of four distinct taxa (C. jusselii Eastw. (Eastwood, 1940), Orthocarpus pilosus S. Wats. (Watson, 1871), C. inconspicua A.Nelson & P.B.Kenn (Nelson & Kennedy, 1906), C. nana Eastw. (Eastwood, 1902a)), two of which (O. pilosus and C. nana) are the type specimens for C. nana and C. pilosa (Fig. 8). The remaining two taxa were later incorporated into C. nana (C. inconspicua) and C. pilosa (C. jusselii), effectively meaning that these entities are no different from C. nana and C. pilosa. However, when we place our best approximation of C. inconspicua in morphospace (i.e., a specimen of the same taxon (C. inconspicua is a synonym of C. nana) that was measured by us that is as geographically close to the type collection of C. inconspicua as possible), we find that this collection occupies a region of morphospace very different from that of the type collection of C. nana (Fig. 8). By including this species into the concept of C. nana through synonymization in the Intermountain Flora (Cronquist et al., 1984), the amount of variation attributed to C. nana likely expanded.

Figure 8 Position of type collections.

Position of type collections of focal taxa and associated synonyms, within the known ranges of each taxon (B) and the corresponding position of the nearest geographic individual that we have measurements for in our dataset is identified in morphospace (A). Individuals are color-coded according to taxonomic identification: Castilleja pilosa var. pilosa (blue), C. pilosa var. longispica (orange), C. pilosa var. steenensis (red), and C. nana (yellow).

Areas of sympatry are not the only source of potential confusion in the taxonomic history of either taxon. For example, the synonymization of C. lapidicola A.Heller (Heller, 1912) in eastern Nevada with C. nana also expanded the region of morphospace attributed to C. nana ((Cronquist et al., 1984), Fig. 8). Similarly, in northern California the inclusion of C. ochracea Eastw. (Eastwood, 1941) and C. pisttacinus (Orthocarpus psittacinus Eastw. (Eastwood, 1902b); C. psittacina (Eastw.) Pennell (Abrams, 1951)) increased the area of morphospace occupied by C. pilosa ((Cronquist et al., 1984); Fig. 8). Ultimately, the qualitative decisions made about species boundaries based on regional treatments have extended and inflated the morphological concepts of both taxa. By going through this procedure of quantifying morphological variation, we can visualize what morphological variation the taxonomy currently embodies. It is apparent that the morphological concept of both C. nana and C. pilosa have expanded through the incorporation of additional taxa as synonyms, and it is possible that the species description of both taxa may no longer represent the features of either taxon.

Conclusion

The inflation of morphological variation attributed to C. nana and C. pilosa during species level revisions, much of them regionally based, in addition to an apparent reliance on potentially plastic morphological characters to distinguish species in sympatry, has resulted in a great deal of morphological confusion in this complex. This likely contributes to the tumultuous taxonomic history of these taxa, and suggests that relying on morphology alone to define species boundaries in this complex is problematic. This is where molecular and ecological lines of evidence will be incredibly important to delimit species. In a robust and integrated delimitation of species, we may find that taxa that have been synonymized are not truly part of their corresponding taxa, or vice versa. Subsequent classifications should reflect these boundaries and highlight the similarities and differences between them.

Here we have begun that process by quantifying morphological variation in this species complex and we have estimated the position of type specimens in that space. The next steps in this group will be to gather molecular and ecological evidence to contribute to a robust species delimitation that is based on multiple lines of evidence. With all data in hand, we can more confidently apply names, whether that is applying an old name, a new name, or combining them all in one.

Supplemental Information

Supplemental Information 1 Primary analyses without outlier removal.

Results of Principal Coordinates Analysis (PCoA) (A) and fuzzy-clustering analyses (B) where no outliers were removed from theanalysis. The PCoA plot shows the first two axes of variation where individuals are plotted in morphospace, and colored according to species identification. Results of fuzzy clustering for k = 4 clusters (left), k = 3 clusters (middle), and k = 2 clusters (right). For each set of silhouettes, the width of each bar corresponds to the silhouette coefficient for that individual in the analysis; average silhouette coefficient for each analysis (k = 4, 3, 2) is reported. Bars are painted with colors corresponding to species identification. In both plots, yellow = C. nana, blue = C. pilosa var. pilosa, orange = C. pilosa var. longispica, red = C. pilosa var. steenensis.

Click here for additional data file.

Supplemental Information 2 Assignment probabilities for fuzzy clustering.

Results of Principal Coordinates Analysis (PCoA; A) and the mapping of assignment probabilities from fuzzy clustering analyses onto points in morphospace (B–D). In each plot, each point is divided into segments corresponding to recovered clusters (identified by different colors). The size of the segment corresponds to the assignment probability of that individual to that cluster—larger segments correspond to higher assignment probabilities and smaller segments correspond to lower assignment probabilities. The purple and green colors only serve to distinguish different clusters; the top left panel (with individuals painted as yellow = C. nana, blue = C. pilosa var. pilosa, orange = C. pilosa var. longispica, and red = C. steenensis) serves as a reference for the taxonomic identification of each individuals.

Click here for additional data file.

Supplemental Information 3 Kernel density estimates.

Kernel density estimates of raw trait values for the continuous traits measured in this study (A-K), organized by species. C. pilosa (including C. pilosa var. longispica, C. pilosa var. pilosa, and C. pilosa var. steenensis) (blue) and C. nana (yellow).

Click here for additional data file.

Supplemental Information 4 Voucher information for collections used in this study.

Information pertaining to herbarium where specimen is housed, accession number of herbarium specimen (when available; some collections are currently in curation), collector and collection number of specimen, latitude and longitude of collection is included for each specimen.

Click here for additional data file.

Supplemental Information 5 Mean and standard deviation of raw, continuous trait values for measured individuals, organized by taxon.

Click here for additional data file.

Supplemental Information 6 Raw counts of categorical traits scored for each individual, organized by taxon.

Click here for additional data file.

Supplemental Information 7 Raw results of fuzzy-clustering analyses for k = 4, 3, and 2 clusters.

For each collection, for each analysis, the cluster assignment and silhouette coefficient are reported. Additionally, for each cluster in each analysis, a membership coefficient is reported for each individual. Finally, the taxonomic identification of each individual is provided.

Click here for additional data file.

Supplemental Information 8 Raw data used in this study.

Click here for additional data file.

Supplemental Information 9 R scripts used in primary analyses in this study.

Click here for additional data file.

The authors would like to thank the following herbaria for providing access to specimens for morphological measurements: College of Idaho Harold M. Tucker Herbarium (CIC), University of Idaho Stillinger Herbarium (ID), University of Montana Herbarium (MONTU), Oregon State University Herbarium (OSC), University of Wyoming Rocky Mountain Herbarium (RM), Washington State University Marion Ownbey Herbarium (WS), University of Washington Herbarium (WTU). Additionally, the authors would like to thank two anonymous reviewers whose thoughtful comments and suggestions were greatly appreciated and contributed to the improvement of the manuscript.

Additional Information and Declarations

Competing Interests

Author Contributions

Data Availability

The authors declare that they have no competing interests.

Sarah J. Jacobs conceived and designed the experiments, performed the experiments, analyzed the data, contributed reagents/materials/analysis tools, prepared figures and/or tables, authored or reviewed drafts of the paper, approved the final draft.

Sarah Herzog conceived and designed the experiments, performed the experiments, authored or reviewed drafts of the paper, approved the final draft.

David C. Tank conceived and designed the experiments, contributed reagents/materials/analysis tools, authored or reviewed drafts of the paper, approved the final draft.

The following information was supplied regarding data availability:

The raw data and analytical scripts used to perform analyses are available in the Supplemental Files.

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
