# Peer review of "Quantifying morphological variation in the Castilleja pilosa species complex (Orobanchaceae)"

_PeerJ, doi:10.7717/peerj.7090_

## Round 0.1 · original submission · Major Revisions

Overall both reviewers agreed that this paper is worth publishing, just take into consideration issues raised by them. The most important are that in Introduction more detail on the specific species complex and not in Castilleja in general is needed. Both reviewers raised concern on the concepts of species delimitation and type specimens, you have to clarify differences and take conclusions based on them. Please upload a new file with your raw data in a format that can be useful in all platforms. I appreciate your careful consideration of the concerns explained by reviewers.

Reviewer 1 ·

Basic reporting

This study thoughtfully quantifies morphological variation in the Castilleja pilosa species complex and considers the consequences of this variation for species delimitation and taxonomy. Integrating taxonomic and morphological data into species delimitation is extremely important and this is an excellent example of how these data can be collected and used. The manuscript is clearly written and methodologically sound, however there are places where I think the writing could be streamlined and places that need further clarification.

The manuscript is well written with an appropriate introduction and sufficient detail. The introduction and discussion should focus more on this specific complex rather than on Castilleja as a whole. For instance are these species annuals? Is there ploidy variation within the complex?

The only problem I encountered with the data is that the file “Jacobs.etal.raw.data.csv” is not readable by either excel or as a text file. I see only jibberish when I open it.

Experimental design

The research question is well defined and the statistical methods used seem reasonable. That said, I think that the authors should consider some of the issues addressed in the recent study of Cadena et al 2018 “Issues and persepctives in species delimitation using phenotypic data.” That paper brings up several topics that are relevant here:
Primary among these is that while PCA and similar reduction of dimensionality techniques find the axes of greatest variation, these are not necessarily the axes that are most useful for distinguishing clusters or species (Cadena et al. 2018). Since all of the axes from the PCoA are used in the subsequent analyses, this point will not change the results, but the authors should be careful in the way they discuss the results of the PCoA. For example lines 304-306 and 360-362 seem to imply that the axes of the greatest morphological distance among individuals are also the axes of greatest distinction among potential clusters.
The second issue that should be discussed is how fuzzy clustering will perform when quantitative traits within species are normally distributed. For example would samples on the tail ends of the trait distribution have lower assignment probabilities?

I am also somewhat concerned by the outlier points that were excluded. It would be helpful to have information in the supplemental material that describes these more thoroughly—perhaps a figure like Figure 7 or replicate analyses that include these outliers?

A summary definition for the silhouette coefficient would be helpful in the caption for Figure 6.

How much clinal variation is there across the ranges of these species? Are there consistent trends for these traits when plotted against latitude?

Line 299-302. How much variation do the first 3 PC axes account for?

Validity of the findings

The data is robust and the conclusions are reasonable.

Additional comments

The figures are very nice and easy to interpret.

Line 267—This assumption seems very questionable to me. Are there any traits that can be measured from the type specimens? Any data to confirm the hypothesized position of the type specimens in the morphospace would be helpful.

Line 65—While I agree that there can be taxonomic drift where the amount of variation ascribed to a certain species can change through time, I don’t agree that a type specimen ought to serve as a “central, anchoring point within the range of known variation for a species.” Type specimens are historical reference points and do not necessarily reflect the most common characteristics of a species. As single individuals sampled from a species, type specimens will necessarily represent only a portion of the variation that exists within the species.

Line 85—specify “young” here with “age”

Line 157—‘known’ is not the right word here because this is something that is being evaluated.

Line 161—what is “peak maturity”? I initially assumed it meant with fruit, but I suspect it means peak flower?

Line 415—I’m not sure that synonomizing is the problem. There do not appear to be morphological clusters within these groups—do you think that there are really clusters within them?

Figure 3. Kernel density plots like this one look really pretty but they also tend to obscure the data—particularly the sample sizes (it looks as though var. steenensis has the same number of individuals as var. longispica). This figure should be replaced with a plot that shows the data more clearly—perhaps a violin plot with points?

Reviewer 2 ·

Basic reporting

see below

Experimental design

no comment

Validity of the findings

see below

Additional comments

Overall, this is a well done study of morphological variation in a small group of Castilleja species. The methods used are appropriate and I think that the conclusions are reasonable.

The main place that I would raise concerns is in the context and description of the nature of species and the meaning of types – basic ideas of taxonomy and nomenclature. First, names have types, taxa (such as species) do not have types, so it is important to distinguish between the two things. The taxonomic act is to delimit or circumscribe the species, while the nomenclatural act is to use types to determine what the correct names for those species are. This paper seems to confuse the two aspects to some extent.

The typical approach to performing a study such as the one presented here is to analyze the variation and decide what patterns of variation are worth recognizing as species. Then one uses types and their position in the variation pattern to decide what the names of the species should be. While it is OK to evaluate previously named taxa in the way that its done here, focusing on the previous circumscriptions so much instead of just looking at the variation pattern (e,g, without colors to represent previous taxa as determined by how they key out in previous treatments) seems to miss the opportunity to evaluate the variation with a fresh and less biased view, asking the question “Are there distinct units within the pattern of variation that are worth recognizing?”

Specific points to raise are the following (and some are noted on the ms. itself):

32-33: Species boundaries are not represented by type specimens – this does not make sense. Circumscriptions are a taxonomic decision, whereas application of types is a nomenclatural act that just follows the rules.

39-40: I don’t think that “Species delimitation relies on our ability to define boundaries between one population and the next.” Is an accurate statement. Species delimitation has been practiced for decades-hundreds of years based on herbarium specimens – and often still is. That does not require knowing anything about population boundaries. Few current studies really deal with the limits of populations.

47-49: this is true but the authors seem to take “species delimitation” as a newly discovered activity, when really that is what has been practiced all along. If they mean “model-based molecular approaches” then it would be best to say that.

54-56: This suggests a misunderstanding of what a type is. Types are not necessarily central points around which the variation of a taxon sweeps. There is no need for them to be central points in variation and certainly for most taxa they are not. A type is just a specimen that fixes the name once a circumscription has been defined. As long as it falls within that variation, the name (if oldest), is the correct one.

64-67: This is partially true but again seems to misunderstand the nature of types (not central points).

104: What does “herbage” mean here? It is not a scientific term – in general use it means “plant stuff” – is that what is meant here – the whole plant?

200: By “nominal categorical” do the authors mean anything different that just “categorical” for characters? I think that the latter is more commonly used.

264-267: Why didn’t you just measure the features on the type specimen and use that instead of this rather indirect approach? If you did this then you would not have to “hypothesize” the placement of the type (line 29), you could just determine it.

405: Taxa or names (such as those given) are not the type specimens for anything. They may be the basionyms if those epithets are the correct ones that are then used in a new combination.

426-427: Again, this is a misunderstanding of what types are.

Annotated reviews are not available for download in order to protect the identity of reviewers who chose to remain anonymous.

---

## Round 0.2 · accepted · Accept

Thank you for taking time to include and consider every comment by the reviewers, it seems that by doing these changes the manuscript improved a lot. I found some typos which you should look to correct while in production. Congratulations again.